# STABLE TARGET FIELD FOR REDUCED VARIANCE SCORE ESTIMATION IN DIFFUSION MODELS

**Yilun Xu**[*], **Shangyuan Tong**[*], **Tommi Jaakkola**
Computer Science and Artificial Intelligence Lab,
Massachusetts Institute of Technology
`ylxu@mit.edu`; `{sytong, tommi}@csail.mit.edu`

## ABSTRACT

Diffusion models generate samples by reversing a fixed forward diffusion process. Despite already providing impressive empirical results, these diffusion models algorithms can be further improved by reducing the variance of the training targets in their denoising score-matching objective. We argue that the source of such variance lies in the handling of intermediate noise-variance scales, where multiple modes in the data affect the direction of reverse paths. We propose to remedy the problem by incorporating a reference batch which we use to calculate weighted conditional scores as more stable training targets. We show that the procedure indeed helps in the challenging intermediate regime by reducing (the trace of) the covariance of training targets. The new stable targets can be seen as trading bias for reduced variance, where the bias vanishes with increasing reference batch size. Empirically, we show that the new objective improves the image quality, stability, and training speed of various popular diffusion models across datasets with both general ODE and SDE solvers. When used in combination with EDM (Karras et al., 2022), our method yields a current SOTA FID of 1.90 with 35 network evaluations on the unconditional CIFAR-10 generation task. The code is available at `https://github.com/Newbeeer/stf`

## 1 INTRODUCTION

Diffusion models (Sohl-Dickstein et al., 2015; Song & Ermon, 2019; Ho et al., 2020) have recently achieved impressive results on a wide spectrum of generative tasks, such as image generation (Nichol et al., 2022; Song et al., 2021b), 3D point cloud generation (Luo & Hu, 2021) and molecular conformer generation (Shi et al., 2021; Xu et al., 2022a). These models can be subsumed under a unified framework in the form of Itô stochastic differential equations (SDE) (Song et al., 2021b). The models learn time-dependent score fields via score-matching (Hyvärinen & Dayan, 2005), which then guides the reverse SDE during generative sampling. Popular instances of diffusion models include variance-exploding (VE) and variance-preserving (VP) SDE (Song et al., 2021b). Building on these formulations, EDM (Karras et al., 2022) provides the best performance to date.

We argue that, despite achieving impressive empirical results, the current training scheme of diffusion models can be further improved. In particular, the variance of training targets in the denoising score-matching (**DSM**) objective can be large and lead to suboptimal performance. To better understand the origin of this instability, we decompose the score field into three regimes. Our analysis shows that the phenomenon arises primarily in the intermediate regime, which is characterized by multiple modes or data points exerting comparable influences on the scores. In other words, in this regime, the sources of the noisy examples generated in the course of the forward process become ambiguous. We illustrate the problem in Figure 1(a), where each stochastic update of the score model is based on disparate targets.

We propose a generalized version of the denoising score-matching objective, termed the *Stable Target Field* (**STF**) objective. The idea is to include an additional *reference batch* of examples that are used to calculate weighted conditional scores as targets. We apply self-normalized importance sampling to aggregate the contribution of each example in the reference batch. Although this process can substantially reduce the variance of training targets (Figure 1(b)), especially in the intermediate regime,

---

[*]Equal Contribution.

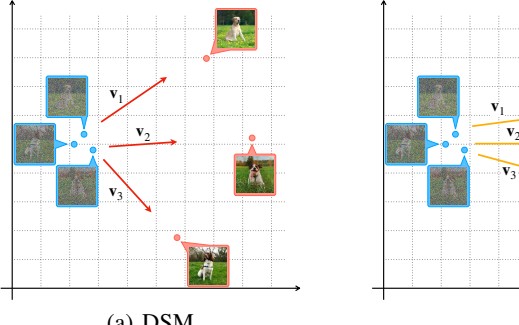

(a) DSM                                          (b) STF

Figure 1: Illustration of differences between the DSM objective and our proposed STF objective. The "destroyed" images (in blue box) are close to each other while their sources (in red box) are not. Although the true score in expectation is the weighted average of $\mathbf{v}_i$, the individual training updates of the DSM objective have a high variance, which our STF objective reduces significantly by including a large reference batch (yellow box).

it does introduce some bias. However, we show that the bias together with the trace-of-covariance of the STF training targets shrinks to zero as we increase the size of the reference batch.

Experimentally, we show that our STF objective achieves new state-of-the-art performance on CIFAR-10 unconditional generation when incorporated into EDM (Karras et al., 2022). The resulting FID score (Heusel et al., 2017) is 1.90 with 35 network evaluations. STF also improves the FID/Inception scores for other variants of score-based models, *i.e.*, VE and VP SDEs (Song et al., 2021b), in most cases. In addition, it enhances the stability of converged score-based models on CIFAR-10 and CelebA $64^2$ across random seeds, and helps avoid generating noisy images in VE. STF accelerates the training of score-based models ($3.6\times$ speed-up for VE on CIFAR-10) while obtaining comparable or better FID scores. To the best of our knowledge, STF is the first technique to accelerate the training process of diffusion models. We further demonstrate the performance gain with increasing reference batch size, highlighting the negative effect of large variance.

Our contributions are summarized as follows: **(1)** We detail the instability of the current diffusion models training objective in a principled and quantitative manner, characterizing a region in the forward process, termed *the intermediate phase*, where the score-learning targets are most variable (Section 3). **(2)** We propose a generalized score-matching objective, *stable target field*, which provides more stable training targets (Section 4). **(3)** We analyze the behavior of the new objective and prove that it is asymptotically unbiased and reduces the trace-of-covariance of the training targets by a factor pertaining to the reference batch size in the intermediate phase under mild conditions (Section 5). **(4)** We illustrate the theoretical arguments empirically and show that the proposed STF objective improves the performance, stability, and training speed of score-based methods. In particular, it achieves the current state-of-the-art FID score on the CIFAR-10 benchmark when combined with EDM (Section 6).

## 2 BACKGROUND ON DIFFUSION MODELS

In diffusion models, the forward process[1] is an SDE with no learned parameter, in the form of:

$$\mathrm{d}\mathbf{x} = \mathbf{f}(\mathbf{x}, t)\mathrm{d}t + g(t)\mathrm{d}\mathbf{w},$$

where $\mathbf{x} \in \mathbb{R}^d$ with $\mathbf{x}(0) \sim p_0$ being the data distribution, $t \in [0, 1]$, $\mathbf{f} \colon \mathbb{R}^d \times [0, 1] \to \mathbb{R}^d$, $g \colon [0, 1] \to \mathbb{R}$, and $\mathbf{w} \in \mathbb{R}^d$ is the standard Wiener process. It gradually transforms the data distribution to a known prior as time goes from 0 to 1. Sampling of diffusion models is done via a corresponding reverse-time SDE (Anderson, 1982):

$$\mathrm{d}\mathbf{x} = \left[\mathbf{f}(\mathbf{x}, t) - g(t)^2 \nabla_{\mathbf{x}} \log p_t(\mathbf{x})\right] \mathrm{d}\bar{t} + g(t)\mathrm{d}\bar{\mathbf{w}},$$

where $\bar{\cdot}$ denotes time traveling backward from 1 to 0. Song et al. (2021b) proposes a probability flow ODE that induces the same marginal distribution $p_t(\mathbf{x})$ as the SDE: $\mathrm{d}\mathbf{x} =$

---

[1]For simplicity, we focus on the version where the diffusion coefficient $g(t)$ is independent of $\mathbf{x}(t)$.

$\left[\mathbf{f}(\mathbf{x}, t) - \frac{1}{2}g(t)^2 \nabla_{\mathbf{x}} \log p_t(\mathbf{x})\right] d\bar{t}$. Both formulations progressively recover $p_0$ from the prior $p_1$. We estimate the score of the transformed data distribution at time $t$, $\nabla_{\mathbf{x}} \log p_t(\mathbf{x})$, via a neural network, $\mathbf{s}_\theta(\mathbf{x}, t)$. Specifically, the training objective is a weighted sum of the denoising score-matching (Vincent, 2011):

$$\min_\theta \ \mathbb{E}_{t \sim q_t(t)} \lambda(t) \mathbb{E}_{\mathbf{x} \sim p_0} \mathbb{E}_{\mathbf{x}(t) \sim p_{t|0}(\cdot|\mathbf{x})} \left[\|\mathbf{s}_\theta(\mathbf{x}(t), t) - \nabla_{\mathbf{x}(t)} \log p_{t|0}(\mathbf{x}(t)|\mathbf{x})\|_2^2\right], \quad (1)$$

where $q_t$ is the distribution for time variable, *e.g.*, $\mathcal{U}[0, 1]$ for VE/VP (Song et al., 2021b) and a log-normal distribution for EDM Karras et al. (2022), and $\lambda(t) = \sigma_t^2$ is the positive weighting function to keep the time-dependent loss at the same magnitude (Song et al., 2021b), and $p_{t|0}(\mathbf{x}(t)|\mathbf{x})$ is the transition kernel denoting the conditional distribution of $\mathbf{x}(t)$ given $\mathbf{x}$[2]. Specifically, diffusion models "destroy" data according to a diffusion process utilizing Gaussian transition kernels, which result in $p_{t|0}(\mathbf{x}(t)|\mathbf{x}) = \mathcal{N}(\boldsymbol{\mu}_t, \sigma_t^2 \boldsymbol{I})$. Recent works (Xu et al., 2022b; Rissanen et al., 2022) have also extended the underlying principle from the diffusion process to more general physical processes where the training objective is not necessarily score-related.

## 3 UNDERSTANDING THE TRAINING TARGET IN SCORE-MATCHING OBJECTIVE

The vanilla denoising score-matching objective at time $t$ is:

$$\ell_{\text{DSM}}(\theta, t) = \mathbb{E}_{p_0(\mathbf{x})} \mathbb{E}_{p_{t|0}(\mathbf{x}(t)|\mathbf{x})}[\|\mathbf{s}_\theta(\mathbf{x}(t), t) - \nabla_{\mathbf{x}(t)} \log p_{t|0}(\mathbf{x}(t)|\mathbf{x})\|_2^2], \quad (2)$$

where the network is trained to fit the individual targets $\nabla_{\mathbf{x}(t)} \log p_{t|0}(\mathbf{x}(t)|\mathbf{x})$ at $(\mathbf{x}(t), t)$ – the "influence" exerted by clean data $\mathbf{x}$ on $\mathbf{x}(t)$. We can swap the order of the sampling process by first sampling $\mathbf{x}(t)$ from $p_t$ and then $\mathbf{x}$ from $p_{0|t}(\cdot|\mathbf{x}(t))$. Thus, $\mathbf{s}_\theta$ has a closed form minimizer:

$$\mathbf{s}_{\text{DSM}}^*(\mathbf{x}(t), t) = \mathbb{E}_{p_{0|t}(\mathbf{x}|\mathbf{x}(t))}[\nabla_{\mathbf{x}(t)} \log p_{t|0}(\mathbf{x}(t)|\mathbf{x})] = \nabla_{\mathbf{x}(t)} \log p_t(\mathbf{x}(t)). \quad (3)$$

The score field is a conditional expectation of $\nabla_{\mathbf{x}(t)} \log p_{t|0}(\mathbf{x}(t)|\mathbf{x})$ with respect to the posterior distribution $p_{0|t}$. In practice, a Monte Carlo estimate of this target can have high variance (Owen, 2013; Elvira & Martino, 2021). In particular, when multiple modes of the data distribution have comparable influences on $\mathbf{x}(t)$, $p_{0|t}(\cdot|\mathbf{x}(t))$ is a multi-mode distribution, as also observed in Xiao et al. (2022). Thus the targets $\nabla_{\mathbf{x}(t)} \log p_{t|0}(\mathbf{x}(t)|\mathbf{x})$ vary considerably across different $\mathbf{x}$ and this can strongly affect the estimated score at $(\mathbf{x}(t), t)$, resulting in slower convergence and worse performance in practical stochastic gradient optimization (Wang et al., 2013).

To quantitatively characterize the variations of individual targets at different time, we propose a metric – the average trace-of-covariance of training targets at time $t$:

$$V_{\text{DSM}}(t) = \mathbb{E}_{p_t(\mathbf{x}(t))} \left[\text{Tr}(\text{Cov}_{p_{0|t}(\mathbf{x}|\mathbf{x}(t))}(\nabla_{\mathbf{x}(t)} \log p_{t|0}(\mathbf{x}(t)|\mathbf{x})))\right]$$

$$= \mathbb{E}_{p_t(\mathbf{x}(t))} \mathbb{E}_{p_{0|t}(\mathbf{x}|\mathbf{x}(t))} \left[\|\nabla_{\mathbf{x}(t)} \log p_{t|0}(\mathbf{x}(t)|\mathbf{x})) - \nabla_{\mathbf{x}(t)} \log p_t(\mathbf{x}(t))\|_2^2\right]. \quad (4)$$

We use $V_{\text{DSM}}(t)$ to define three successive phases relating to the behavior of training targets. As shown in Figure 2(a), the three phases partition the score field into near, intermediate, and far regimes (Phase 1~3 respectively). Intuitively, $V_{\text{DSM}}(t)$ peaks in the intermediate phase (Phase 2), where multiple distant modes in the data distribution have comparable influences on the same noisy perturbations, resulting in unstable targets. In Phase 1, the posterior $p_{0|t}$ concentrates around one single mode, thus low variation. In Phase 3, the targets remain similar across modes since $\lim_{t \to 1} p_{t|0}(\mathbf{x}(t)|\mathbf{x}) \approx p_1$ for commonly used transition kernels.

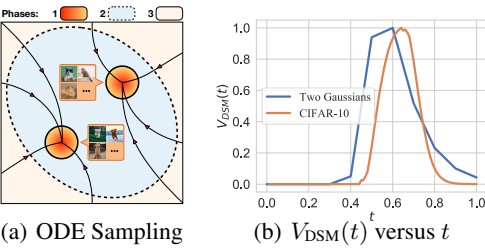

(a) ODE Sampling  (b) $V_{\text{DSM}}(t)$ versus $t$

Figure 2: **(a)**: Illustration of the three phases in a two-mode distribution. **(b)**: Estimated $V_{\text{DSM}}(t)$ for two distributions. We normalize the maximum value to 1 for illustration purposes.

We validate this argument empirically in Figure 2(b), which shows the estimated $V_{\text{DSM}}(t)$ for a mixture of two Gaussians as well as a subset of CIFAR-10 dataset (Krizhevsky et al., 2009) for a more

---

[2]We omit "(0)" from $\mathbf{x}(0)$ when there is no ambiguity.

realistic setting. Here we use VE SDE, *i.e.*, $p_{t|0}(\mathbf{x}(t)|\mathbf{x}) = \mathcal{N}\left(\mathbf{x}, \sigma_m^2(\frac{\sigma_M}{\sigma_m})^{2t}\mathbf{I}\right)$ for some $\sigma_m$ and $\sigma_M$ (Song et al., 2021b). $V_{\text{DSM}}(t)$ exhibits similar phase behavior across $t$ in both toy and realistic cases. Moreover, $V_{\text{DSM}}(t)$ reaches its maximum value in the intermediate phase, demonstrating the large variations of individual targets. We defer more details to Appendix C.

## 4 TREATING SCORE AS A FIELD

The vanilla denoising score-matching approach (Equation 3) can be viewed as a Monte Carlo estimator, *i.e.*, $\nabla_{\mathbf{x}(t)} \log p_t(\mathbf{x}(t)) = \mathbb{E}_{p_{0|t}(\mathbf{x}|\mathbf{x}(t))}[\nabla_{\mathbf{x}(t)} \log p_{t|0}(\mathbf{x}(t)|\mathbf{x})] \approx \frac{1}{n}\sum_{i=1}^n \nabla_{\mathbf{x}(t)} \log p_{t|0}(\mathbf{x}(t)|\mathbf{x}_i)$ where $\mathbf{x}_i$ is sampled from $p_{0|t}(\cdot|\mathbf{x}(t))$ and $n = 1$. The variance of a Monte Carlo estimator is proportional to $\frac{1}{n}$, so we propose to use a larger batch ($n$) to counter the high variance problem described in Section 3. Since sampling directly from the posterior $p_{0|t}$ is not practical, we first apply importance sampling with the proposal distribution $p_0$. Specifically, we sample a large reference batch $\mathcal{B}_L = \{\mathbf{x}_i\}_{i=1}^n \sim p_0^n$ and get the following approximation:

$$\nabla_{\mathbf{x}(t)} \log p_t(\mathbf{x}(t)) \approx \frac{1}{n}\sum_{i=1}^n \frac{p_{0|t}(\mathbf{x}_i|\mathbf{x}(t))}{p_0(\mathbf{x}_i)} \nabla_{\mathbf{x}(t)} \log p_{t|0}(\mathbf{x}(t)|\mathbf{x}_i).$$

The importance weights can be rewritten as $p_{0|t}(\mathbf{x}|\mathbf{x}(t))/p_0(\mathbf{x}) = p_{t|0}(\mathbf{x}(t)|\mathbf{x})/p_t(\mathbf{x}(t))$. However, this basic importance sampling estimator has two issues. The weights now involve an unknown normalization factor $p_t(\mathbf{x}(t))$ and the ratio between the prior and posterior distribution can be large in high dimensional spaces. To remedy these problems, we appeal to self-normalization techniques (Hesterberg, 1995) to further stabilize the training targets:

$$\nabla_{\mathbf{x}(t)} \log p_t(\mathbf{x}(t)) \approx \sum_{i=1}^n \frac{p_{t|0}(\mathbf{x}(t)|\mathbf{x}_i)}{\sum_{j=1}^n p_{t|0}(\mathbf{x}(t)|\mathbf{x}_j)} \nabla_{\mathbf{x}(t)} \log p_{t|0}(\mathbf{x}(t)|\mathbf{x}_i). \tag{5}$$

We term this new training target in Equation 5 as *Stable Target Field (STF)*. In practice, we sample the reference batch $\mathcal{B}_L = \{\mathbf{x}_i\}_{i=1}^n$ from $p_0^n$ and obtain $\mathbf{x}(t)$ by applying the transition kernel to the "first" training data $\mathbf{x}_1$. Taken together, the new STF objective becomes:

$$\ell_{\text{STF}}(\theta, t) = \mathbb{E}_{\{\mathbf{x}_i\}_{i=1}^n \sim p_0^n} \mathbb{E}_{\mathbf{x}(t) \sim p_{t|0}(\cdot|\mathbf{x}_1)}$$
$$\left[\left\|\mathbf{s}_\theta(\mathbf{x}(t), t) - \sum_{k=1}^n \frac{p_{t|0}(\mathbf{x}(t)|\mathbf{x}_k)}{\sum_{j=1}^n p_{t|0}(\mathbf{x}(t)|\mathbf{x}_j)} \nabla_{\mathbf{x}(t)} \log p_{t|0}(\mathbf{x}(t)|\mathbf{x}_k)\right\|_2^2\right]. \tag{6}$$

When $n = 1$, STF reduces to the vanilla denoising score-matching (Equation 2). When $n > 1$, STF incorporates a reference batch to stabilize training targets. Intuitively, the new weighted target assigns larger weights to clean data with higher influence on $\mathbf{x}(t)$, *i.e.*, higher transition probability $p_{t|0}(\mathbf{x}(t)|\mathbf{x})$.

Similar to our analysis in Section 3, we can again swap the sampling process in Equation 6 so that, for a perturbation $\mathbf{x}(t)$, we sample the reference batch $\mathcal{B}_L = \{\mathbf{x}_i\}_{i=1}^n$ from $p_{0|t}(\cdot|\mathbf{x}(t))p_0^{n-1}$, where the first element involves the posterior, and the rest follow the data distribution. Thus, the minimizer of the new objective (Equation 6) is (derivation can be found in Appendix B.1)

$$\mathbf{s}_{\text{STF}}^*(\mathbf{x}(t), t) = \mathbb{E}_{\mathbf{x}_1 \sim p_{0|t}(\cdot|\mathbf{x}(t))} \mathbb{E}_{\{\mathbf{x}_i\}_{i=2}^n \sim p_0^{n-1}} \left[\sum_{k=1}^n \frac{p_{t|0}(\mathbf{x}(t)|\mathbf{x}_k)}{\sum_j p_{t|0}(\mathbf{x}(t)|\mathbf{x}_j)} \nabla_{\mathbf{x}(t)} \log p_{t|0}(\mathbf{x}(t)|\mathbf{x}_k)\right]. \tag{7}$$

Note that although STF significantly reduces the variance, it introduces bias: the minimizer is no longer the true score. Nevertheless, in Section 5, we show that the bias converges to 0 as $n \to \infty$, while reducing the trace-of-covariance of the training targets by a factor of $n$ when $p_{0|t} \approx p_0$. We further instantiate the STF objective (Equation 6) with transition kernels in the form of $p_{t|0}(\mathbf{x}(t)|\mathbf{x}) = \mathcal{N}(\mathbf{x}, \sigma_t^2\mathbf{I})$, which includes EDM (Karras et al., 2022), VP (through reparameterization) and VE (Song et al., 2021b):

$$\mathbb{E}_{\mathbf{x}_1 \sim p_{0|t}(\cdot|\mathbf{x}(t))} \mathbb{E}_{\{\mathbf{x}_i\}_{i=2}^n \sim p_0^{n-1}} \left[\left\|\mathbf{s}_\theta(\mathbf{x}(t), t) - \frac{1}{\sigma_t^2}\sum_{k=1}^n \frac{\exp\left(-\frac{\|\mathbf{x}(t)-\mathbf{x}_k\|_2^2}{2\sigma_t^2}\right)}{\sum_j \exp\left(-\frac{\|\mathbf{x}(t)-\mathbf{x}_j\|_2^2}{2\sigma_t^2}\right)}(\mathbf{x}_k - \mathbf{x}(t))\right\|_2^2\right].$$

To aggregate the time-dependent STF objective over $t$, we sample the time variable $t$ from the training distribution $q_t$ and apply the weighting function $\lambda(t)$. Together, the final training objective for STF is $\mathbb{E}_{t \sim q_t(t)} [\lambda(t) \ell_{\text{STF}}(\theta, t)]$. We summarize the training process in Algorithm 1. The small batch size $|\mathcal{B}|$ is the same as the normal batch size in the vanilla training process. We defer specific use cases of STF objectives combined with various popular diffusion models to Appendix A.

---

**Algorithm 1** Learning the stable target field

---

**Input:** Training iteration $T$, Initial model $\mathbf{s}_\theta$, dataset $\mathcal{D}$, learning rate $\eta$.
**for** $t = 1 \dots T$ **do**
    Sample a large reference batch $\mathcal{B}_L$ from $\mathcal{D}$, and subsample a small batch $\mathcal{B} = \{\mathbf{x}_i\}_{i=1}^{|\mathcal{B}|}$ from $\mathcal{B}_L$
    Uniformly sample the time $\{t_i\}_{i=1}^{|\mathcal{B}|} \sim q_t(t)^{|\mathcal{B}|}$
    Obtain the batch of perturbed samples $\{\mathbf{x}_i(t_i)\}_{i=1}^{|\mathcal{B}|}$ by applying the transition kernel $p_{t|0}$ on $\mathcal{B}$
    Calculate the stable target field of $\mathcal{B}_L$ for all $\mathbf{x}_i(t_i)$:
$$\mathbf{v}_{\mathcal{B}_L}(\mathbf{x}_i(t_i)) = \sum_{\mathbf{x} \in \mathcal{B}_L} \frac{p_{t_i|0}(\mathbf{x}_i(t_i)|\mathbf{x})}{\sum_{\mathbf{y} \in \mathcal{B}_L} p_{t_i|0}(\mathbf{x}_i(t_i)|\mathbf{y})} \nabla_{\mathbf{x}_i(t_i)} \log p_{t_i|0}(\mathbf{x}_i(t_i)|\mathbf{x})$$
    Calculate the loss: $\mathcal{L}(\theta) = \frac{1}{|\mathcal{B}|} \sum_{i=1}^{|\mathcal{B}|} \lambda(t_i) \|\mathbf{s}_\theta(\mathbf{x}_i(t_i), t_i) - \mathbf{v}_{\mathcal{B}_L}(\mathbf{x}_i(t_i))\|_2^2$
    Update the model parameter: $\theta = \theta - \eta \nabla \mathcal{L}(\theta)$
**end for**
**return** $\mathbf{s}_\theta$

---

## 5 ANALYSIS

In this section, we analyze the theoretical properties of our approach. In particular, we show that the new minimizer $\mathbf{s}_{\text{STF}}^*(\mathbf{x}(t), t)$ (Equation 7) converges to the true score asymptotically (Section 5.1). Then, we show that the proposed STF reduces the trace-of-covariance of training targets propositional to the reference batch size in the intermediate phase, with mild conditions (Section 5.2).

### 5.1 ASYMPTOTIC BEHAVIOR

Although in general $\mathbf{s}_{\text{STF}}^*(\mathbf{x}(t), t) \neq \nabla_{\mathbf{x}(t)} \log p_t(\mathbf{x}(t))$, the bias shrinks toward 0 with a increasing $n$. In the following theorem we show that the minimizer of STF objective at $(\mathbf{x}(t), t)$, i.e., $\mathbf{s}_{\text{STF}}^*(\mathbf{x}(t), t)$, is asymptotically normal when $n \to \infty$.

**Theorem 1.** *Suppose* $\forall t \in [0, 1], 0 < \sigma_t < \infty$, *then*

$$\sqrt{n} \left( \mathbf{s}_{STF}^*(\mathbf{x}(t), t) - \nabla_{\mathbf{x}(t)} \log p_t(\mathbf{x}(t)) \right) \xrightarrow{d} \mathcal{N} \left( \mathbf{0}, \frac{\text{Cov}(\nabla_{\mathbf{x}(t)} p_{t|0}(\mathbf{x}(t)|\mathbf{x}))}{p_t(\mathbf{x}(t))^2} \right) \quad (8)$$

We defer the proof to Appendix B.2. The theorem states that, for commonly used transition kernels, $\mathbf{s}_{\text{STF}}^*(\mathbf{x}(t), t) - \nabla_{\mathbf{x}(t)} \log p_t(\mathbf{x}(t))$ converges to a zero mean normal, and larger reference batch size ($n$) will lead to smaller asymptotic variance. As can be seen in Equation 8, when $n \to \infty$, $\mathbf{s}_{\text{STF}}^*(\mathbf{x}(t), t)$ highly concentrates around the true score $\nabla_{\mathbf{x}(t)} \log p_t(\mathbf{x}(t))$.

### 5.2 TRACE OF COVARIANCE

We now highlight the small variations of the training targets in the STF objective compared to the DSM. As done in Section 3, we study the trace-of-covariance of training targets in STF:

$$V_{\text{STF}}(t) = \mathbb{E}_{p_t(\mathbf{x}(t))} \left[ \text{Tr} \left( \text{Cov}_{p_{0|t}(\cdot|\mathbf{x}(t)) p_0^{n-1}} \left( \sum_{k=1}^{n} \frac{p_{t|0}(\mathbf{x}(t)|\mathbf{x}_k)}{\sum_j p_{t|0}(\mathbf{x}(t)|\mathbf{x}_j)} \nabla_{\mathbf{x}(t)} \log p_{t|0}(\mathbf{x}(t)|\mathbf{x}_k) \right) \right) \right].$$

In the following theorem we compare $V_{\text{STF}}$ with $V_{\text{DSM}}$. In particular, we can upper bound $V_{\text{STF}}(t)$ by

**Theorem 2.** *Suppose* $\forall t \in [0, 1], 0 < \sigma_t < \infty$, *then*

$$V_{STF}(t) \leq \frac{1}{n-1} \left( V_{DSM}(t) + \frac{\sqrt{3}d}{\sigma_t^2} \sqrt{\mathbb{E}_{p_t(\mathbf{x}(t))} D_f \left( p_0(\mathbf{x}) \| p_{0|t}(\mathbf{x}|\mathbf{x}(t)) \right)} \right) + O \left( \frac{1}{n^2} \right),$$

*where $D_f$ is an f-divergence with $f(y) = \begin{cases} (1/y - 1)^2 & (y < 1.5) \\ 8y/27 - 1/3 & (y \geq 1.5) \end{cases}$. Further, when $n \gg d$ and*

$p_{0|t}(\mathbf{x}|\mathbf{x}(t)) \approx p_0(\mathbf{x})$ *for all* $\mathbf{x}(t)$, $V_{STF}(t) \lesssim \frac{V_{DSM}(t)}{n-1}$.

We defer the proof to Appendix B.3. The second term that involves $f$-divergence $D_f$ is necessary to capture how the coefficients, *i.e.*, $p_{t|0}(\mathbf{x}(t)|\mathbf{x}_k)/\sum_j p_{t|0}(\mathbf{x}(t)|\mathbf{x}_j)$ used to calculate the weighted score target, vary across different samples $\mathbf{x}(t)$. This term decreases monotonically as a function of $t$. In Phase 1, $p_{0|t}(\mathbf{x}|\mathbf{x}(t))$ differs substantially from $p_0(\mathbf{x})$ and the divergence term $D_f$ dominates. In contrast to the upper bound, both $V_{STF}(t)$ and $V_{DSM}(t)$ have minimal variance at small values of $t$ since the training target is always dominated by one $\mathbf{x}$. The theorem has more relevance in Phase 2, where the divergence term decreases to a value comparable to $V_{DSM}(t)$. In this phase, we empirically observe that the ratio of the two terms in the upper bound ranges from 10 to 100. Thus, when we use a large reference batch size (in thousands), the theorem implies that STF offers a considerably lower variance (by a factor of 10 or more) relative to the DSM objective. In Phase 3, the second term vanishes to 0, as $p_t \approx p_{t|0}$ with large $\sigma_t$ for commonly used transition kernels. As a result, STF reduces the average trace-of-covariance of the training targets by at least $n-1$ times in the far field.

Together, we demonstrate that the STF targets have diminishing bias (Theorem 1) and are much more stable during training (Theorem 2). These properties make the STF objective more favorable for diffusion models training with stochastic gradient optimization.

## 6 EXPERIMENTS

In this section, we first empirically validate our theoretical analysis in Section 5, especially for variance reduction in the intermediate phase (Section 6.1). Next, we show that the STF objective improves various diffusion models on image generation tasks in terms of image quality (Section 6.2). In particular, STF achieves state-of-the-art performance on top of EDM. In addition, we demonstrate that STF accelerates the training of diffusion models (Section 6.3), and improves the convergence speed and final performance with an increasing reference batch size (Section 6.3).

### 6.1 VARIANCE REDUCTION IN THE INTERMEDIATE PHASE

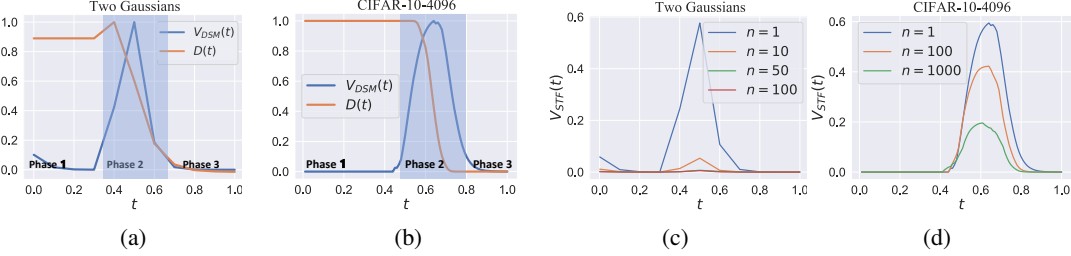

(a)        (b)        (c)        (d)

Figure 3: **(a, b)**: $V_{DSM}(t)$ and $D(t)$ versus $t$. We normalize the maximum values to 1 for illustration purposes. **(c, d)**: $V_{STF}(t)$ with a varying reference batch size $n$.

The proposed Algorithm 1 utilizes a large reference batch to calculate the stable target field instead of the individual target. In addition to the theoretical analysis in Section 5, we provide further empirical study to characterize the intermediate phase and verify the variance reduction effects by STF. Apart from $V(t)$, we also quantify the average divergence between the posterior $p_{0|t}(\cdot|\mathbf{x}(t))$ and the data distribution $p_0$ at time $t$ (introduced in Theorem 2): $D(t) = \mathbb{E}_{p_t(\mathbf{x}(t))} \left[ D_f \left( p_{0|t}(\mathbf{x}|\mathbf{x}(t)) \parallel p_0(\mathbf{x}) \right) \right]$. Intuitively, the number of high-density modes in $p_{0|t}(\cdot|\mathbf{x}(t))$ grows as $D(t)$ decreases. To investigate their behaviors, we construct two synthetic datasets: (1) a 64-dimensional mixture of two Gaussian components (**Two Gaussians**), and (2) a subset of 1024 images of CIFAR-10 (**CIFAR-10-4096**).

Figure 3(a) and Figure 3(b) show the behaviors of $V_{DSM}(t)$ and $D(t)$ on Two Gaussian and CIFAR-10-4096. In both settings, $V_{DSM}(t)$ reaches its peak in the intermediate phase (Phase 2), while $D(t)$ gradually decreases over time. These results agree with our theoretical understanding from Section 3. In Phase 2 and 3, several modes of the data distribution have noticeable influences on the scores, but only in Phase 2 are the influences much more distinct, leading to high variations of the individual target $\nabla_{\mathbf{x}(t)} \log p_{t|0}(\mathbf{x}(t)|\mathbf{x})$, $\mathbf{x} \sim p_{0|t}(\cdot|\mathbf{x}(t))$.

Figure 3(c) and Figure 3(d) further show the relationship between $V_{\text{STF}}(t)$ and the reference batch size $n$. Recall that when $n = 1$, STF degenerates to individual target and $V_{\text{STF}}(t) = V_{\text{DSM}}(t)$. We observe that $V_{\text{STF}}(t)$ decreases when enlarging $n$. In particular, the predicted relation $V_{\text{STF}}(t) \lessgtr V_{\text{DSM}}(t)/(n-1)$ in Theorem 2 holds for the two Gaussian datasets where $D_f$ is small. On the high dimensional dataset CIFAR-10-4096, the stable target field can still greatly reduce the training target variance with large reference batch sizes $n$.

## 6.2 IMAGE GENERATION

Table 1: CIFAR-10 sample quality (FID, Inception) and number of function evaluation (NFE).

| Methods | Inception ↑ | FID ↓ | NFE ↓ |
|---|---|---|---|
| StyleGAN2-ADA (Karras et al., 2020) | 9.83 | 2.92 | 1 |
| DDPM (Ho et al., 2020) | 9.46 | 3.17 | 1000 |
| NCSNv2 (Song & Ermon, 2020) | 8.40 | 10.87 | 1161 |
| PFGM (Xu et al., 2022b) | 9.68 | 2.48 | 104 |
| ***VE** (Song et al., 2021b)* | | | |
| DSM - RK45 | 9.27 | 8.90 | 264 |
| STF (ours) - RK45 | 9.52 ↑ | 5.51 ↓ | 200 |
| DSM - PC | 9.68 | 2.75 | 2000 |
| STF (ours) - PC | 9.86 ↑ | 2.66 ↓ | 2000 |
| ***VP** (Song et al., 2021b)* | | | |
| DSM - DDIM | 9.20 | 5.16 | 100 |
| STF (ours) - DDIM | 9.28 ↑ | 5.06 ↓ | 100 |
| DSM - RK45 | 9.46 | 2.90 | 140 |
| STF (ours) - RK45 | 9.43 ↓ | 2.99 ↑ | 140 |
| ***EDM** (Karras et al., 2022)* | | | |
| DSM - Heun, NCSN++ | 9.82 | 1.98 | 35 |
| STF (ours) - Heun, NCSN++ | **9.93** ↑ | **1.90** ↓ | 35 |
| DSM - Heun, DDPM++ | 9.78 | 1.97 | 35 |
| STF (ours) - Heun, DDPM++ | 9.79 ↑ | 1.92 ↓ | 35 |

We demonstrate the effectiveness of the new objective on image generation tasks. We consider CIFAR-10 (Krizhevsky et al., 2009) and CelebA $64 \times 64$ (Yang et al., 2015) datasets. We set the reference batch size $n$ to 4096 (CIFAR-10) and 1024 (CelebA $64^2$). We choose the current state-of-the-art score-based method EDM (Karras et al., 2022) as the baseline, and replace the DSM objective with our STF objective during training. We also apply STF to two other popular diffusion models, VE/VP SDEs (Song et al., 2021b). For a fair comparison, we directly adopt the architectures and the hyper-parameters in Karras et al. (2018) and Song et al. (2021b) for EDM and VE/VP respectively. In particular, we use the improved NCSN++/DDPM++ models (Karras et al., 2022) in the EDM scheme. To highlight the stability issue, we train three models with different seeds for VE on CIFAR-10. We provide more experimental details in Appendix D.1.

**Numerical Solver.** The reverse-time ODE and SDE in scored-based models are compatible with any general-purpose solvers. We use the adaptive solver RK45 method (Dormand & Prince, 1980; Song et al., 2021b) (RK45) for VE/VP and the popular DDIM solver (Song et al., 2021a) for VP. We adopt Heun's 2nd order method (Heun) and the time discretization proposed by Karras et al. (2022) for EDM. For SDEs, we apply the predictor-corrector (PC) sampler used in (Song et al., 2021b). We denote the methods in a objective-sampler format, *i.e.*, **A-B**, where $\mathbf{A} \in \{\text{DSM, STF}\}$ and $\mathbf{B} \in \{\text{RK45, PC, DDIM, Heun}\}$. We defer more details to Appendix D.2.

**Results.** For quantitative evaluation of the generated samples, we report the FID scores (Heusel et al., 2017) (lower is better) and Inception (Salimans et al., 2016) (higher is better). We measure the sampling speed by the average NFE (number of function evaluations). We also include the results of several popular generative models (Karras et al., 2020; Ho et al., 2020; Song & Ermon, 2019; Xu et al., 2022b) for reference.

Table 1 and Table 2 report the sample quality and the sampling speed on unconditional generation of CIFAR-10 and CelebA $64^2$. Our main findings are: **(1) STF achieves new state-of-the-art FID scores for unconditional generation on CIFAR-10 benchmark.** As shown in Ta-

ble 1, The STF objective obtains a FID of 1.90 when incorporated with the EDM scheme. To the best of our knowledge, this is the lowest FID score on the unconditional CIFAR-10 generation task. In addition, the STF objective consistently improves the EDM across the two architectures. **(2) The STF objective improves the performance of different diffusion models.** We observe that the STF objective improves the FID/Inception scores of VE/VP/EDM on CIFAR-10, for most ODE and SDE samplers. STF consistently provides performance gains for VE across datasets. Remarkably, our objective achieves much better sample quality using ODE samplers for VE, with an FID score gain of 3.39 on CIFAR-10, and 2.22 on Celeba $64^2$.

For VP, STF provides better results on the popular DDIM sampler, while suffering from a slight performance drop when using the RK45 sampler. **(3) The STF objective stabilizes the converged VE model with the RK45 sampler.** In Appendix E.1, we report the standard deviations of performance metrics for converged models with different seeds on CIFAR-10 with VE. We observe that models trained with the STF objective give more consistent results, with a smaller standard deviation of used metrics.

We further provide generated samples in Appendix F. One interesting observation is that when using the RK45 sampler for VE on CIFAR-10, the generated samples from the STF objective do not contain noisy images, unlike the vanilla DSM objective.

Table 2: FID and NFE on CelebA $64^2$

| Methods/NFEs | FID ↓ | NFE ↓ |
|---|---|---|
| *CelebA $64^2$ - RK45* | | |
| VE (DSM) | 7.56 | 260 |
| VE (STF) | **5.34** | 266 |
| *CelebA $64^2$ - PC* | | |
| VE (DSM) | 9.13 | 2000 |
| VE (STF) | **8.28** | 2000 |

## 6.3 ACCELERATING TRAINING OF DIFFUSION MODELS

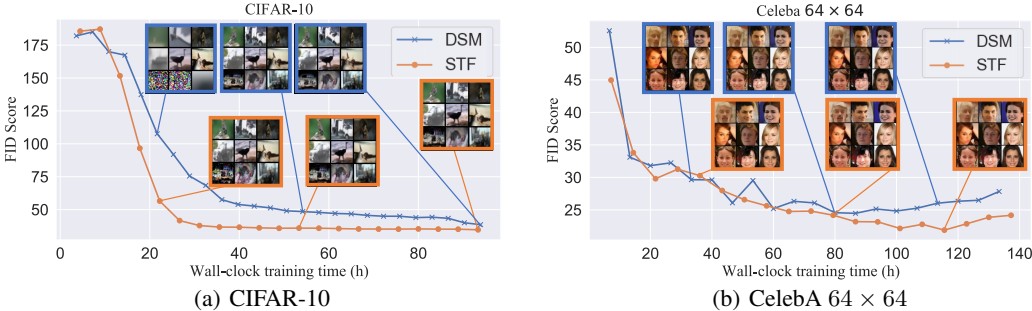

(a) CIFAR-10                (b) CelebA $64 \times 64$

Figure 4: FID and generated samples throughout training on (a) CIFAR-10 and (b) CelebA $64^2$.

The variance-reduction techniques in neural network training can help to find better optima and achieve faster convergence rate (Wang et al., 2013; Defazio et al., 2014; Johnson & Zhang, 2013). In Figure 4, we demonstrate the FID scores every 50k iterations during the course of training. Since our goal is to investigate relative performance during the training process, and because the FID scores computed on 1k samples are strongly correlated with the full FID scores on 50k sample (Song & Ermon, 2020), we report FID scores on 1k samples for faster evaluations. We apply ODE samplers for FID evaluation, and measure the training time on two NVIDIA A100 GPUs. For a fair comparison, we report the average FID scores of models trained by the DSM and STF objective on VE versus the wall-clock training time (h).

The STF objective achieves better FID scores with the same training time, although the calculation of the target field by the reference batch introduces slight overhead (Algorithm 1). In Figure 4(a), we show that the STF objective drastically accelerates the training of diffusion models on CIFAR-10. The STF objective achieves comparable FID scores with $3.6\times$ less training time (25h versus 90h). For CelebA $64^2$ datasets, the training time improvement is less significant than on CIFAR-10. Our hypothesis is that the STF objective is more effective when there are multiple well-separated modes in data distribution, *e.g.*, the ten classes in CIFAR-10, where the DSM objective suffer from relatively larger variations in the intermediate phase. In addition, the converged models have better final performance when pairing with the STF on both datasets.

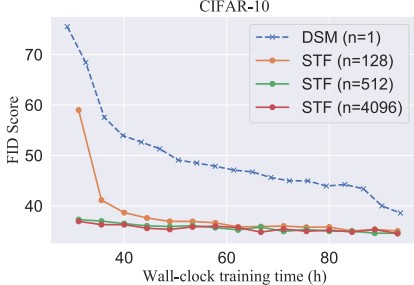

Figure 5: FID scores in the training course with varying reference batch size.

### 6.4 EFFECTS OF THE REFERENCE BATCH SIZE

According to our theory (Theorem 2), the upper bound of the trace-of-covariance of the STF target decreases proportionally to the reference batch size. Here we study the effects of the reference batch size $(n)$ on model performances during training. The FID scores are evaluated on $1k$ samples using the RK45 sampler. As shown in Figure 5, models converge faster and produce better samples when increasing $n$. It suggests that smaller variations of the training targets can indeed speed up training and improve the final performances of diffusion models.

## 7 RELATED WORK

**Different phases of diffusion models.** The idea of diffusion models having different phases has been explored in prior works though the motivations and definitions vary (Karras et al., 2022; Choi et al., 2022). Karras et al. (2022) argues that the training targets are difficult and unnecessary to learn in the very near field (small $t$ in our Phase 1), whereas the training targets are always dissimilar to the true targets in the intermediate and far field (our Phase 2 and Phase 3). As a result, their solution is sampling $t$ with a log-normal distribution to emphasize the relevant region (relatively large $t$ in our Phase 1). In contrast, we focus on reducing large training target variance in the intermediate and far field, and propose STF to better estimate the true target (cf. Karras et al. (2022)). Choi et al. (2022) identifies a key region where the model learns perceptually rich contents, and determines the training weights $\lambda(t)$ based on the signal-to-noise ratio (SNR) at different $t$. As SNR is monotonically decreasing over time, the resulting up-weighted region does not match our Phase 2 characterization. In general, our proposed STF method reduces the training target variance in the intermediate field and is complementary to previous improvements of diffusion models.

**Importance sampling.** The technique of importance sampling has been widely adopted in machine learning community, such as debiasing generative models (Grover et al., 2019), counterfactual learning (Swaminathan & Joachims, 2015) and reinforcement learning (Metelli et al., 2018). Prior works using importance sampling to improve generative model training include reweighted wake-sleep (RWS) (Bornschein & Bengio, 2014) and importance weighted autoencoders (IWAE) (Burda et al., 2015). RWS views the original wake-sleep algorithm (Hinton et al., 1995) as importance sampling with one latent variable, and proposes to sample multiple latents to obtain gradient estimates with lower bias and variance. IWAE utilizes importance sampling with multiple latents to achieve greater flexibility of encoder training and tighter log-likelihood lower bound compared to the standard variational autoencoder (Kingma & Welling, 2013; Rezende et al., 2014).

**Variance reduction for Fisher divergence.** One popular approach to score-matching is to minimize the Fisher divergence between true and predicted scores (Hyvärinen & Dayan, 2005). Wang et al. (2020) links the Fisher divergence to denoising score-matching (Vincent, 2011) and studies the large variance problem (in $O(1/\sigma_t^4)$) of the Fisher divergence when $t \to 0$. They utilize a control variate to reduce the variance. However, this is typically not a concern for current diffusion models as the time-dependent objective can be viewed as multiplying the Fisher divergence by $\lambda(t) = \sigma_t^2$, resulting in a finite-variance objective even when $t \to 0$.

## 8 CONCLUSION

We identify large target variance as a significant training issue affecting diffusion models. We define three phases with distinct behaviors, and show that the high-variance targets appear in the intermediate phase. As a remedy, we present a generalized score-matching objective, *Stable Target Field* (STF), whose formulation is analogous to the self-normalized importance sampling via a large reference batch. Albeit no longer an unbiased estimator, our proposed objective is asymptotically unbiased and reduces the trace-of-covariance of the training targets, which we demonstrate theoretically and empirically. We show the effectiveness of our method on image generation tasks, and show that STF improves the performance, stability, and training speed over various state-of-the-art diffusion models. Future directions include a principled study on the effect of different reference batch sampling procedures. Our presented approach is uniformly sampling from the whole dataset $\{\mathbf{x}_i\}_{i=2}^n \sim p_0^{n-1}$, so we expect that training diffusion models with a reference batch of more samples in the neighborhood of $\mathbf{x}_1$ (the sample from which $\mathbf{x}(t)$ is perturbed) would lead to an even better estimation of the score field. Moreover, the three-phase analysis can effectively capture the behaviors of other physics-inspired generative models, such as PFGM (Xu et al., 2022b) or the more advanced PFGM++ (Xu et al., 2023). Therefore, we anticipate that STF can enhance the performance and stability of these models further.

ACKNOWLEDGEMENTS

We are grateful to Benson Chen for reviewing an early draft of this paper. We would like to thank Hao He and the anonymous reviewers for their valuable feedback. YX and TJ acknowledge support from MIT-DSTA Singapore collaboration, from NSF Expeditions grant (award 1918839) "Understanding the World Through Code", and from MIT-IBM Grand Challenge project. ST and TJ also acknowledge support from the ML for Pharmaceutical Discovery and Synthesis Consortium (MLPDS).

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
