# OpenReview forum: "Stable Target Field for Reduced Variance Score Estimation in Diffusion Models"
_ICLR.cc/2023/Conference — ICLR 2023 poster_

### Official Review · Reviewer_Qtdt · 2022-10-24

**Confidence:** 3
**Correctness:** 3
**Technical Novelty And Significance:** 3
**Empirical Novelty And Significance:** 2
**Recommendation:** 5

**Clarity, Quality, Novelty And Reproducibility:**

The paper is clear, has good writing quality and novelty. Source code was provided for reproducibility.

**Strength And Weaknesses:**

**Strengths**
- I found the paper interesting to read, with the teorethical
  part well  explained.
- The paper demonstrates in simple examples the high target variance
  of vanilla score-matching, and how it is reduced when using the
  proposed approach.
- The approach improves the sampling of the baseline methods, without
  incurring in extra training time. In the case of CIFAR-10, it
  significantly improves the training efficiency.

**Weaknesses**

- The empirical evidence in favor of the method is rather weak, with
  experiments showing some improvements only for the small CIFAR-10
  benchmark.
- In Figures 6 and 7, FID is computed on only 1K samples, yet the
  Inception features are typically of higher dimension, so the Gaussian
  approximation for the FID would be ill-conditioned. I wonder if the
  authors can comment on that (Curves show scores of ~40, while ~2 is
  reported in Table 1.)


**Summary Of The Paper:**

This paper proposes a method to reduce the sample variance for
training score-matching models. The method is based on using multiple
reference samples and weighted importance sampling to compute more
stable targets. Small-scale experiments show improvement when using
the proposed scheme.


**Summary Of The Review:**

The paper proposes an interesting idea for improving score-matching training. On the flip side the paper presents limited
experimental evidence in favor of the proposed approach.

---

> ### Author Response · Authors · 2022-11-14
> **Thank you for your review and suggestions**
>
> Thank you for the detailed review and thoughtful feedback. Below we address specific questions.
>
> **Q: The empirical evidence in favor of the method is rather weak, with experiments showing some improvements only for the small CIFAR-10 benchmark.**
>
> A: We have conducted experiments on two higher-resolution datasets, CelebA $64^2$ and ImageNet $64^2$. It shows that our STF objective can still improve over the vanilla objective for VE-SDE on these two datasets. Specifically, STF improves the FID scores on CelebA $64^2$/ImageNet $64^2$ unconditional generation task from $9.13/25.96$ to $8.28/24.33$, when using the PC sampler. We have included these results in Table 2 in our revised version.
>
> We also note that CIFAR-10 is the most commonly used benchmark for generative models and has served as the main benchmark in most advanced score-based/diffusion models [1,2,3]. Although the size of CIFAR-10 is relatively small compared to ImageNet (50k versus 1.28M), the multi-modal nature of CIFAR-10 is still challenging for generative modeling. In our revised version, we demonstrate that STF achieves the new state-of-the-art performance on CIFAR-10 with an FID score of $1.90$ when STF is combined with the EDM scheme [1].
>
> *[1] Tero Karras, Miika Aittala, Timo Aila, and Samuli Laine. Elucidating the Design Space of Diffusion-Based Generative Models. NeurIPS 2022.*
>
> *[2] Yang Song, Jascha Sohl-Dickstein, Diederik P. Kingma, Abhishek Kumar, Stefano Ermon, and Ben Poole. Score-Based Generative Modeling through Stochastic Differential Equations. ICLR 2021.*
>
> *[3] Jonathan Ho, Ajay Jain, and Pieter Abbeel. Denoising diffusion probabilistic models. NeurIPS 2020.*
>
> **Q: In Figures 6 and 7, FID is computed on only 1K samples, yet the Inception features are typically of higher dimension, so the Gaussian approximation for the FID would be ill-conditioned. I wonder if the authors can comment on that (Curves show scores of ~40, while ~2 is reported in Table 1.)**
>
> A: Thanks for raising this question, and it is indeed important to clarify. We added an explanation in Section 6.3. Our goal is to investigate relative performance during the training process in Section 6.3. The relative value of FID scores computed on 1k samples serves as a good proxy for the relative value of the full FID scores on 50k samples, as also observed and used in [1]. Hence we choose to report FID scores on 1k samples for faster evaluations across checkpoints.
>
> As most previous works compare the full FID scores on 50k samples across methods, we report the full FID score in Table 1 when comparing to baselines and other reference methods on CIFAR-10.
>
> *[1] Yang Song and Stefano Ermon. Improved Techniques for Training Score-Based Generative Models, NeurIPS 2020*

---

### Official Review · Reviewer_B7Es · 2022-10-26

**Confidence:** 4
**Correctness:** 4
**Technical Novelty And Significance:** 4
**Empirical Novelty And Significance:** 3
**Recommendation:** 8

**Clarity, Quality, Novelty And Reproducibility:**

This paper is of high quality with an original observation, theoretically sound fix, and sufficient theoretical and empirical evidence to support the claim.

The technical part of this paper is very well-written. It clearly states the variance problem, and introduces all necessary backgrounds to understand the fix. Nevertheless, I feel the introduction part can be improved. It frequently mentions terminology like stable targets, direction of inverse path, reference batch, etc. It is hard to get the idea by just looking at the first two pages. I suggest to revise them and point out the variance problem more directly.

Minor:

* At the beginning of section 4: "The vanilla denoising score-matching approach (Equation 3) can viewed as", missing a verb


**Strength And Weaknesses:**

## Strengths

* I find the paper very interesting, address a relevant problem in score-based model training, and potentially have high impact. Simply observing the variance problem in the middle stage of training is nontrivial and requires a deep understanding of the denoising score matching idea and the training of score-based models.

* Moreover, the proposed fix is simple and very practical. Both the theory and empirical results suggest the modified loss has lower variance.

## Weaknesses

I do not see much weakness except a few minor points below:

* It would be nice to give a derivation of eq. (7) in appendix.
* It would be nice to see if there a connection between the modified denoising score matching loss and an importance-weighted version of the ELBO. See, e.g.,

Bornschein, J., & Bengio, Y. (2014). Reweighted wake-sleep. arXiv preprint arXiv:1406.2751.
Burda, Y., Grosse, R., & Salakhutdinov, R. (2015). Importance weighted autoencoders. arXiv preprint arXiv:1509.00519.

* since this paper focus on reducing variance of score matching loss, the following paper that proposes a control variate for DSM could be relevant

Wang, Z., Cheng, S., Yueru, L., Zhu, J., & Zhang, B. (2020, June). A wasserstein minimum velocity approach to learning unnormalized models. In International Conference on Artificial Intelligence and Statistics (pp. 3728-3738). PMLR.

**Summary Of The Paper:**

Score-based generative models are learned via denoising score matching at many time steps during a diffusion process. This paper identifies an important problem with such loss function---estimating this loss via simple monte carlo could have high variance in the middle stage of the diffusion. The authors then propose to use self-normalized importance sampling to reduce the variance, which leads to a new objective function called stable target field. Empirical results show that this new objective consistently improves the performance of score-based generative models.

**Summary Of The Review:**

I find this paper identify an important problem, propose a nice solution, and is sufficiently practical to have large impact. Therefore, I recommend acceptance.

---

> ### Author Response · Authors · 2022-11-14
> **Thank you for your review and suggestions**
>
> Thank you for the detailed review and thoughtful feedback. Below we address specific questions.
>
>
> **Q: It would be nice to give a derivation of eq. (7) in Appendix.**
>
> A: Thanks for the suggestion. We have included a detailed derivation of Equation 7 in Appendix B.1.
>
> **Q: It would be nice to see if there is a connection between the modified denoising score matching loss and an importance-weighted version of the ELBO. See, e.g., [1, 2].**
>
> A: Thanks for the pointers. We have expanded our related work discussion on importance sampling to include these two papers in Section 7, second paragraph. They are related in the sense that we all adopt importance sampling (with normalization) to reduce the variance of training objectives. Specifically, all three methods (reweighted wake-sleep, RWS [1]; importance weighted autoencoder, IWAE [2]; our STF) are generalizations of the original formulations (wake-sleep [3], VAE [4], and SGMs respectively). However, specific use cases are somewhat different. RWS uses importance sampling for parameter gradient estimates while IWAE uses importance sampling for greater flexibility of encoder training and to get a tighter bound on the log-likelihood.
>
> *[1] Jörg Bornschein and Yoshua Bengio. Reweighted wake-sleep. arXiv preprint arXiv:1406.2751, 2014.*
>
> *[2] Yuri Burda, Roger Grosse, and Ruslan Salakhutdinov. Importance weighted autoencoders. arXiv preprint arXiv:1509.00519, 2015.*
>
> *[3] Geoffrey E Hinton, Peter Dayan, Brendan J Frey, and Radford M Neal. The “wake-sleep” algorithm for unsupervised neural networks. Science, 268(5214):1158–1161, 1995.*
>
> *[4] Diederik P Kingma and Max Welling. Auto-encoding variational bayes. arXiv preprint arXiv:1312.6114, 2013.*
>
> **Q: Since this paper focus on reducing variance of score matching loss, the paper [1] that proposes a control variate for DSM could be relevant.**
>
> A: Thanks for the reference. It is an interesting work on reducing the variance of denoising score matching (DSM). We have added a detailed discussion in Section 7, paragraph 3. They link DSM to Fisher divergence and study the large variance problem (in $O(1/\sigma_t^4)$) of the Fisher divergence estimator when the added noise is very small ($t\to0$, meaning $\sigma_t\to0$). They then utilize a control variate to reduce the variance. However, the issue identified in the paper [1] is typically not a concern for the current SGMs training, as the time-dependent objective can be viewed as multiplying the Fisher divergence by $\lambda(t) = \sigma_t^2$. This results in a finite-variance objective even when $t\to0$. In contrast, the variance issue that we address in the current paper deals with training targets at moderately large t (Phase 2).
>
> *[1] Ziyu Wang, Shuyu Cheng, Li Yueru, Jun Zhu, and Bo Zhang. A wasserstein minimum velocity approach to learning unnormalized models. In International Conference on Artificial Intelligence and Statistics, pp. 3728–3738. PMLR, 2020.*
>
> **Q: I feel the introduction part can be improved. It frequently mentions terminology like stable targets, direction of inverse path, reference batch, etc. It is hard to get the idea by just looking at the first two pages. I suggest to revise them and point out the variance problem more directly.**
>
> A: Thanks for the suggestion! We have revised the abstract and introduction accordingly to make the presentation of the variance issue more direct and clear. With this change, the whole paper is also modified slightly to match the new presentation flow. Please let us know if the presentation can be further improved.
>
> Thank you for the corrections for the typos. We have polished the writing according to your suggestions.

---

### Official Review · Reviewer_M5mL · 2022-10-29

**Confidence:** 4
**Correctness:** 3
**Technical Novelty And Significance:** 4
**Empirical Novelty And Significance:** 2
**Recommendation:** 6

**Clarity, Quality, Novelty And Reproducibility:**

**Clarity:** I think the paper is clearly written and was easy to follow for me, despite some fairly technical content.

**Quality:** Overall, the quality of the paper is okay. The paper is well written and motivated with interesting ideas and insights. Most experiments are appropriate, although insufficient. There are some concerns regarding the thoroughness of the experiments. Also, there is very relevant work that is not properly discussed and taken into account (see above).

**Novelty:** Technically, the method clearly is novel, to the best of my knowledge. It is creative and original work. I do have some concerns regarding the relevance and significance, though (see above; benefits only shown clearly for VESDE + ODE sampling).

**Reproducibility:** I think the approach is reproducible. The method is described clearly, experiment details seem to be provided in the appendix, and the submission also includes code.

**Strength And Weaknesses:**

**Strengths:**

- The idea to use an additional reference batch to create a more stable and lower variance target score estimate for denoising score matching is novel. It is an interesting, well-motivated and smart idea.

- I like how the paper derives the STF approach and constructs its main method. The analysis of the original denoising score matching objective, followed by the introduction of the reference batch leveraging an importance sampling framework, and then resorting to self-normalized importance sampling to avoid the intractable normalization factor, is elegant.

- The work includes explicit technical analyses on the bias and variance of the proposed STF objective.

- The variance reduction analyses (Figure 3) in the different phases are quite interesting and insightful.

**Weaknesses:**

- It is nice that the paper provides some analytical analyses on the bias and variance of the STF approach. However, how tight is the bound on the STF variance in Theorem 2? I agree that when $p_{0|t}(x|x(t))\approx p_0(x)$, then the bound is meaningful. But this only really happens in "phase 3" (essentially, phase 3 is defined by $p_{0|t}(x|x(t))\approx p_0(x)$). But the variance reduction also works well in the more relevant "phase 2", as empirically shown. But in this relevant regime, where $D_f$ is not yet 0, the bound is very hard to interpret and provides little insight.

- The paper states that Variance Exploding SDEs (VESDE) are the state-of-the-art SDEs used for training score-based generative diffusion models. It is correct that Song et al. [1] achieved strong results with the VESDE when combined with predictor-corrector SDE sampling. However, almost the entire literature on diffusion models uses Variance Preserving SDEs (VPSDE). Moreover, it has recently been shown how to subsume all different common diffusion model SDEs under one umbrella in Karras et al. [2]. This paper then shows how a VESDE-based model can be improved significantly simply by using appropriate model and network parametrizations and preconditionings (their "EDM" scheme). Consequently, I would consider this EDM the state-of-the-art approach. Moreover, the main question is, does the STF approach also help when applied to more modern and widely used diffusion models, this is, (a) VPSDE and (b) Karras et al.'s EDM approach? The paper focuses on the vanilla VESDE with ODE-based sampling, even though it is well-known that the vanilla VESDE does not play well with ODE solvers and isn't widely used like that in the literature. In fact, already when sampling stochastically with predictor-corrector sampling, the benefit by STF is very marginal and almost not significant (Table 1, PC sampler (SDE)). Also note that the Karras et al. paper is not cited or discussed at all.

- Note that Karras et al. [2] also discuss different perturbation regimes (see paragraph "Loss weighting and sampling" in their Section 5), similar to the different phases discussed in this work. The fact that there are effectively different phases during SGM training is known. This should be acknowledged.

- In Section 3, the paper points to the multimodality of the distribution $p_{0|t}(x|x(t))$. Note that this is extensively discussed also in Xiao et al. [3]. It would make sense to acknowledge that work.

- Regarding "The STF objective stabilizes the model performance at convergence.": I think this is overclaiming. The model only shows a significantly lower variance on the evaluation metric for the setup VESDE + RK45 (already when using the SDE framework with a PC sampler, the variance isn't much lower; Table 1). To verify this claim more broadly, again, experiments on other diffusion models should be run (VPSDE, EDM).

- The paper mentions the computational overhead induced by the reference batch. How small or big is this overhead actually? My understanding is that it is small, because the samples from the reference batch are only used to estimate the target score, but we do not have to call the expensive neural network on them. However, what about memory? Some more details would be helpful here. Note that this comment did not influence my rating significantly.

[1] Song et al., Score-Based Generative Modeling through Stochastic Differential Equations, 2021.

[2] Karras et al., Elucidating the Design Space of Diffusion-Based Generative Models, 2022.

[3] Xiao et al., Tackling the Generative Learning Trilemma with Denoising Diffusion GANs, 2022.

**Summary Of The Paper:**

This paper analyzes the denoising score matching objective used to train diffusion models, which have recently become the dominating class of deep generative models. The paper argues that at intermediate perturbation levels the denoising score matching objective is particularly noisy and "unstable", because one perturbed data sample could correspond to many different original clean data samples (the paper calls this phase 2; at the same time, we aren't yet in the regime where every data point is almost entirely destroyed and the ground truth score is effectively almost the same standard normal score for all data points -- phase 3 in the language of the paper). Standard denoising score matching uses only a single data sample among the plausible ones to estimate the score in each iteration. This effectively leads to high variance training (although unbiased). The paper therefore proposes to include a reference batch of additional samples to calculate a more stable score estimate. This corresponds to a form of importance sampling, introducing an unknown normalization factor. This problem is circumvented with self-normalized importance sampling, which, however, introduces bias. Ultimately, the paper proposes "Stable Target Field" (STF), a biased but lower variance score estimator for training score-based generative models.

Experimentally, the paper shows superior performance of "Variance-Exploding" SDE models when trained with STF compared to standard denoising score matching training. Also, training tends to converge quicker. These advantages primarily manifest when using ODE samplers to generate data.

**Summary Of The Review:**

In summary, I think the main idea and the proposed method of the paper, the STF for training diffusion models, is interesting, well-motivated, and also novel. I do not have any concerns with respect to the technical and methodological novelty.

However, my concern is that the method has been primarily validated on experiments using a standard VESDE scheme, together with ODE solvers for sampling. This setup is known to be challenging and in practice, almost all diffusion models actually rely on a different setup, either the VPSDE or the recent EDM scheme. It isn't clear whether there is any benefit when applied to these methods. If there isn't, then the new method is not particularly important or significant. Consequently, I cannot recommend the paper for publication at this point. I would recommend the authors to run experiments with different SDEs and setups and demonstrate the benefit of STF more broadly.

---

> ### Author Response · Authors · 2022-11-14
> **Thank you for your review and suggestions**
>
> Thank you for the detailed review and thoughtful feedback. In particular, we appreciate the suggestion to incorporate our STF into the state-of-the-art EDM [1] scheme. Our experimental results now show that STF in combination with EDM achieves a new state-of-the-art FID score ($\textbf{1.90}$) on the CIFAR-10 dataset. Below we address your specific questions.
>
> **Q: How tight is the bound on the STF variance in Theorem 2? The variance reduction also works well in the more relevant "phase 2", as empirically shown. But in this relevant regime, where $D_f$ is not yet 0, the bound is very hard to interpret and provides little insight.**
>
> A: Thanks for the question. The bound indeed depends on the phase. In Phase 3 the reduction is guaranteed to be at least on the order of the reference batch size ($n$). In Phase 1, $p_{0|t}(x|x(t))$ differs substantially from $p_0(x)$ and the divergence term $D_f$ dominates. Phase 2 is an interesting case as it balances a factor of ($n-1$) reduction against the divergence. The divergence term is necessary for the bound as it accounts for the variability of the coefficients/weights $\frac{p_{t|0}(x|x_k)}{\sum_{j} p_{t|0}(x|x_j)}$ that multiply the scores in the likelihood weighting step. The additional term in the bound beyond $V_v(t)$ is nevertheless of comparable value in Phase 2. We empirically observe that the ratio of the additional term (the term that involves $D_f$) to $V_v$ – additional term / $V_v$ –  ranges roughly from 10 and 100. In practice, we use a large reference batch where $n$ is in thousands. As a result, the STF objective, even if not reducing variance by a factor of $n$, still provides a considerable reduction. To quantify the reduction more precisely we conduct experiments in Section 6.1 that confirm the above claims. The remaining parts of Section 6 show that STF consistently outperforms various baselines.
>
> **Q: Does the STF approach also help when applied to more modern and widely used diffusion models, this is, (a) VPSDE and (b) Karras et al. [1] 's EDM approach?**
>
>
> A: Thanks for the suggestion! We directly apply our STF objective to the EDM scheme [1]. Specifically, we use the same hyper-parameters, network architectures, preconditioning, loss weighting as in [1] and adopt the same improved NSCN++ architecture. We show that STF improves the FID/Inception scores of the EDM baseline from $1.98/9.82$ to $1.90/9.93$ with 35 NFE on the CIFAR-10 unconditional generation task. We believe this FID score of $1.90$ is now the current state-of-the-art result in the literature. In addition, the STF objective also improves the performance of the EDM baseline when using the improved DDPM++ architecture [1]. We have included the results in Table 1.
>
> The experimental results do suggest that STF improvement is complementary to previous updates and can indeed help more advanced diffusion models as well.
>
> We also briefly tested STF together with the VP-SDE scheme. The results in this case are mixed. STF provides better results on the popular DDIM sampler while suffering from a slight performance drop when using the RK45 sampler.
>
> **Q: Karras et al. [1] also discuss different perturbation regimes (see paragraph "Loss weighing and sampling" in their Section 5), similar to the different phases discussed in this work. The fact that there are effectively different phases during SGM training is known. This should be acknowledged.**
>
> A: Thanks for the suggestion. We have included a detailed discussion on the topic in the restructured related work section (now Section 7, first paragraph). Indeed, prior works have discussed different phases in the forward process. However, the reasoning about phases is different. They argue that training targets are difficult and unnecessary to learn in the very near field (small $t$ in our Phase 1) and always differ substantially from the true targets scores in the intermediate and far field (our Phase 2 and Phase 3). Their solution was to change the sampling of $t$ so as to emphasize the relevant region (relatively large $t$ in our Phase 1). In contrast, we introduce and identify the phases specifically based on the training target variance (especially in Phase 2). Our proposed STF method is complementary to their analysis and also compatible to be used in combination to potentially offer additional gains. Indeed, when incorporated into their framework, we show that STF provides further improvements for SGMs' training.
>
>
> *[1] Tero Karras, Miika Aittala, Timo Aila, and Samuli Laine. Elucidating the Design Space of Diffusion-Based Generative Models. NeurIPS 2022.*

---

> > ### Author Response · Authors · 2022-11-14
> > **Response Part II**
> >
> > **Q: In Section 3, the paper points to the multimodality of the distribution $p_{0|t}(x|x(t))$. Note that this is extensively discussed also in Xiao et al. It would make sense to acknowledge that work.**
> >
> > A: Thanks for the pointer. We agree this is related to our statements about the multi-modality of $p_{0|t}(x|x(t))$ and have included the suggested reference in the relevant section (Section 3).
> >
> >
> > **Q: Regarding "The STF objective stabilizes the model performance at convergence.": I think this is overclaiming. The model only shows a significantly lower variance on the evaluation metric for the setup VESDE + RK45 (already when using the SDE framework with a PC sampler, the variance isn't much lower; Table 1). To verify this claim more broadly, again, experiments on other diffusion models should be run (VPSDE, EDM).**
> >
> > A: Thanks for pointing this out. We have clarified the statement and it now reads "The STF objective stabilizes the converged VE model with the RK45 sampler." in Section 6.2. To measure the stability of converged models, we need to repeat the same experiment several times. We are unable to finish these large-scale experiments for other models during the rebuttal period due to limited time and computational resources. We will update the paper once the results become available.
> >
> > **Q: The paper mentions the computational overhead induced by the reference batch. How small or big is this overhead actually? My understanding is that it is small, because the samples from the reference batch are only used to estimate the target score, but we do not have to call the expensive neural network on them. However, what about memory? Some more details would be helpful here. Note that this comment did not influence my rating significantly.**
> >
> > A: Thanks for asking. We report the actual overhead of the STF and vanilla objectives on the EDM scheme. Our implementation is based on the official GitHub repo of EDM (https://github.com/NVlabs/edm), which adopts the `torch.distributed` framework for distributed GPU training. All the numbers are measured on two NVIDIA A100 GPUs. The table below summarizes the per-GPU memory and per 50k images training time when setting the mini-batch size to 512 and the reference batch size to 1024.
> >
> > |  | Per-GPU memory (G) |  Per 50k images wall-clock time (s)|
> > | --- | :-----------: | :---: |
> > | Vanilla EDM | 36.05 | 98.5 |
> > | STF + EDM | 40.64 | 101.5 |

---

> > > ### Comment · Reviewer_M5mL · 2022-11-22
> > > **Thank you for the response**
> > >
> > > I would like to thank the authors for their detailed reply to my review. Many of my questions have been addressed and the manuscript has been improved. I appreciate in particular the additional experiments for EDM, VP-SDE, etc. Some of the improvements are a bit marginal, but overall these experiments provide a more thorough and representative picture of the method. And methodologically, the proposed method is certainly interesting. Therefore, I decided to raise my score by one point and am now leaning towards suggesting acceptance. I do not have any further questions.

---

### Author Response · Authors · 2022-11-14
**A summary of updates**

We would like to thank all reviewers for their constructive feedback. We have revised our paper according to these comments. Major revisions are highlighted in blue in the new version. In particular, we show that the proposed STF method achieves new state-of-the-art results when incorporated into the EDM [1] scheme (FID score $\textbf{1.90}$ on CIFAR-10). We have also included other new experiments and more discussions of related works. Below we provide a brief summary of these updates

## 1. Experiments on the state-of-the-art diffusion scheme EDM
As suggested by reviewer M5mL, we apply STF to the EDM scheme [1], which is the current state-of-the-art model. We show experimentally that STF achieves a record FID score of $1.90$ on unconditional CIFAR-10 generation with 35 network evaluations,  when using the improved NCSN++ architecture and sampler in [1]. To the best of our knowledge, this is the lowest FID score on this benchmark. In addition, the STF objective consistently improves EDM across architectures, e,g., decreasing the FID scores from $\textbf{1.97/1.98}$ to $\textbf{1.92/1.90}$ with DDPM++/NCSN++ architectures.

## 2. More datasets / diffusion schemes
In response to reviewer Qtdt, we have included experiments on two higher resolution datasets – CelebA $64^2$ and ImageNet $64^2$ – with VE-SDE, to showcase the effectiveness of STF (Table 2).  As suggested by reviewer M5mL, we have applied the STF objective to the VP-SDE scheme in addition to EDM (Table 1).

## 3. More discussions of related works
As suggested by reviewer M5mL and reviewer B7Es, we have included more discussion about related works in Section 7. For example, we discussed relevant notions of phases in score-based models, importance sampling for VAE, and a variance-reduction technique for Fisher divergence.

*[1] Tero Karras, Miika Aittala, Timo Aila, and Samuli Laine. Elucidating the Design Space of Diffusion-Based Generative Models. NeurIPS 2022.*

---

### Decision · Program_Chairs · 2023-01-20

**Decision:**

Accept: poster

**Justification For Why Not Higher Score:**

Albeit the nice theoretical variance reduction results, the empirical results are marginal and whether the proposed technique will be widely used is questionable.

**Justification For Why Not Lower Score:**

The submission provides valuable insights into variance reduction in diffusion models.

**Metareview: Summary, Strengths And Weaknesses:**

It is known that learning score-based generative models via denoising score matching can suffer from high variance training objectives. This work circumvents this problem by introducing a low-variance yet biased training objective, incorporating an importance-weighted denoising objective. The introduced objective is well-motivated and it is analyzed theoretically and empirically in different settings. The experimental results show that the proposed approach can marginally improve the performance of score-based diffusion models. The rebuttal successfully addressed some of the minor issues raised by reviewers. Given this, I am happy to recommend accept.

**Note From Pc:**

if the above contains the word "oral" or "spotlight" please see: "oral" presentation means -> notable-top-5% and "spotlight" means -> notable-top-25%. As stated in our emails, we are disassociating presentation type from AC recommendations

**Summary Of Ac-Reviewer Meeting:**

N/A